# Autoantibodies as Biomarker and Therapeutic Target in Systemic Sclerosis

**DOI:** 10.3390/biomedicines10092150

**Published:** 2022-09-01

**Authors:** Hanna Graßhoff, Konstantinos Fourlakis, Sara Comdühr, Gabriela Riemekasten

**Affiliations:** Department of Rheumatology and Clinical Immunology, University of Lübeck, 23538 Lübeck, Germany

**Keywords:** systemic sclerosis, autoreactive B cells, autoantibodies, rituximab, aptamers, Seldeg

## Abstract

Systemic sclerosis (SSc) is a rare connective tissue disorder characterized by immune dysregulation evoking the pathophysiological triad of inflammation, fibrosis and vasculopathy. In SSc, several alterations in the B-cell compartment have been described, leading to polyclonal B-cell hyperreactivity, hypergammaglobulinemia and autoantibody production. Autoreactive B cells and autoantibodies promote and maintain pathologic mechanisms. In addition, autoantibodies in SSc are important biomarkers for predicting clinical phenotype and disease progression. Autoreactive B cells and autoantibodies represent potentially promising targets for therapeutic approaches including B-cell-targeting therapies, as well as strategies for unselective and selective removal of autoantibodies. In this review, we present mechanisms of the innate immune system leading to the generation of autoantibodies, alterations of the B-cell compartment in SSc, autoantibodies as biomarkers and autoantibody-mediated pathologies in SSc as well as potential therapeutic approaches to target these.

## 1. Introduction

Systemic sclerosis (SSc) is a rare connective tissue disorder characterized by immune dysregulation evoking the pathophysiological triad of inflammation, fibrosis and vasculopathy. This pathophysiological triad results in a heterogenous disease course involving the skin and internal organs such as the lung, the gastrointestinal tract, the heart, or the kidneys.

Several genes have been identified increasing disease susceptibility and several susceptibility haplotypes show an association to defined autoimmune profiles [1,2]. However, exposure to environmental agents and infectious pathogens is thought to play a major role in disease development and maintenance [3,4]. Furthermore, there are hints that the exposure to certain environmental factors leads to distinct clinical phenotypes and possibly also to a distinct autoantibody profile [5,6]. According to clinical and laboratory characteristics, SSc is divided into the limited cutaneous form (lcSSc) and the diffuse cutaneous form (dcSSc). In addition to this classification, further disease subsets have been distinguished, e.g., SSc-overlap syndromes, SSc sine scleroderma or paraneoplastic SSc. Beyond these classifications, however, the disease expression, the course of the disease, as well as the development of secondary disease complications and the mortality of the individual patient are heterogeneous. This heterogeneity has been a major obstacle in performing and analyzing clinical trials in SSc. For example, the risk of death varies greatly depending on the organ manifestations. Therefore, stratification of patients is essential in the therapy of SSc to predict disease progression and response to therapy.

In many other autoimmune diseases, but also in SSc, autoantibodies could be identified, which can be useful for stratification. These autoantibodies are not only valuable biomarkers, but together with autoreactive B cells, they are a crucial hallmark in the pathogenesis of SSc. Therefore, this review aims to address whether autoantibodies can be used as predictors of disease course and which therapies targeting autoantibody-mediated pathologies have been evaluated so far in SSc.

Thus, this review focuses on the following three questions:Which mechanisms are involved in maturation of autoreactive B cells and secretion of autoantibodies in SSc?Which autoantibodies might be useful as predictors of disease course and which role do autoantibodies play in the pathophysiology of SSc?Which therapeutic approaches have been evaluated to target autoantibody-mediated pathologies in SSc?

By highlighting the topic of autoreactive B cells and autoantibodies in SSc, we present a disease-overarching pathomechanism mediating pathologies in several diseases and discuss potential therapeutic approaches [7].

## 2. Which Mechanisms Are Involved in Maturation of Autoreactive B Cells and Autoantibody Secretion in SSc?

B cells represent key cellular players in the pathophysiology of SSc. Accordingly, a gene expression study on SSc skin revealed an increase in gene clusters characteristic of B cells and plasma cells [8]. These data were confirmed by a recently published study also proving a B-cell characteristic signature by RNA sequencing in skin samples of SSc patients [9]. Moreover, CD20+ B-cell infiltrates were demonstrated in pulmonary tissue samples of SSc patients with interstitial lung disease [10].

In general, the B-cell compartment, in which short-lived plasmablasts and long-lived plasma cells secrete antibodies, is affected by a dichotomy between rapid and effective immune defense against pathogens and potentially harmful autoimmunity. To balance this dichotomy, maturation of the B-cell compartment and antibody secretion must be tightly regulated. Divergent regulation of these mechanisms is, therefore, a central feature of autoimmune diseases. In SSc, disease-specific autoantibodies are thought not only to play a role in disease maintenance but also to be involved in the development of the disease, thus representing an early pathomechanism. Evidence of this is that disease-specific autoantibodies in SSc are present before early clinical symptoms such as morning stiffness, Raynaud’s phenomenon or swollen fingers [11]. Moreover, in patients with Raynaud’s phenomenon, detection of autoantibodies predicts microvascular damage in nailfold capillary microscopy and subsequent diagnosis of SSc [12]. This aspect is taken into account in the criteria for “Very Early Diagnosis Of SSc” (VEDOSS), which include the presence of autoantibodies [13].

To understand the potential role of autoantibodies as biomarkers and in pathophysiology, as well as corresponding therapeutic approaches, we present the mechanisms involved in the generation of autoreactive B cells and the secretion of autoantibodies in SSc in the first section.

### 2.1. Tolerance Mechanisms in B-Cell Maturation

In humans, B cells are classified into three subclasses, namely B-1 cells, originating mainly from the fetal liver, B-2 cells, developing in the bone marrow, and regulatory B cells which prohibit the expansion of pro-inflammatory lymphocytes and, thus, contribute to immune homeostasis [14]. Maturation of the B-2 cells in the bone marrow is characterized by the formation of a functional B-cell receptor (BCR). In 2003, Wardemann et al. estimated that 55–75% of early immature B cells in the bone marrow display autoreactivity [15]. To ensure the balance between rapid and effective immune defense and autoimmunity, B cells undergo tolerance mechanisms during maturation (Figure 1).

In the bone marrow, high-affinity binding of endogenous antigens to the BCR of early immature B cells evokes receptor editing, induction of anergy or clonal deletion by apoptosis [16]. These mechanisms are comprised under the term ”central tolerance mechanisms” and reduce frequency of autoreactive B cells from approximately 75% to 43% [15].

Subsequently, late immature B cells are released from the bone marrow to the blood stream. Peripheral tolerance mechanisms take effect between the transitional stages 1 and 2 in the spleen and secondary lymphoid organs. Peripheral tolerance mechanisms comprise B-cell-intrinsic mechanisms such as anergy or clonal deletion and B-cell-extrinsic mechanisms including ignorance and limited secretion of survival factors. These peripheral tolerance mechanisms reduce frequency of autoreactive B cells from approximately 42% to 20% [15].

Subsequently, mature B cells without antigen contact—also called naïve B cells—act as antigen-presenting cells and circulate in the blood and lymphatic organs. Here, the binding of an antigen to the BCR evokes a T-cell-independent or T-cell-dependent activation of the B cell. T-cell-independent activated B cells secrete antibodies of the IgM isotype and do not undergo class switch or formation of memory cells. T-cell-dependent activation of B cells promotes differentiation to plasma cells or to memory B cells. Therefore, a third and fourth important tolerance mechanism controlling activation of autoreactive B cells are the central and peripheral tolerance mechanisms of T cells, as these are more stringent than for B cells and most antigens induce a T-cell-dependent B-cell activation. Moreover, B cells require interaction with T cells for the germinal center reaction with class switch and somatic hypermutation [17].

Further mechanisms that can activate cells of the B-cell compartment are via Toll-like receptors (TLRs). TLRs belong to the innate immune system. With regard to their ligands, a distinction is made between pathogen-associated molecular patterns (PAMPs) and damage-associated molecular patterns (DAMPs). Whereas PAMPs are molecular motifs conserved within a class of microbes, DAMPs comprise molecules released from damaged cells due to trauma or infection. The expression of TLRs varies in the B-cell compartment. Naïve human B cells express only low levels of TLRs, whereas activated and memory B cells express increased levels of TLRs [18]. Moreover, increased gene expression of TLRs occurs physiologically in B cells after stimulation of BCR or CD40 [19]. Stimulation of TLRs on plasma cells has been shown to increase antibody secretion [20].

In addition, dendritic cells express TLRs and, subsequently, these cells could activate the B-cell compartment [21]. Regarding the role of TLRs in SSc, we refer to reviews by O’Reilly [22,23] as well as Frasca and Lande [24]. To date, the role of B-cell activation and the respective effects on antibody secretion via TLRs in the development and maintenance of SSc is poorly understood.

### 2.2. Natural and Pathogenic Autoantibodies

Despite the described tolerance mechanisms, autoreactive B cells and autoantibodies can be detected in the peripheral blood of humans and mice. Though there is a potentially increased risk for autoimmune diseases, in most cases, these diseases do not occur [15,25,26].

In fact, non-harmful autoantibodies represent a substantial proportion of antibodies [27]. This group of antibodies is termed natural autoantibodies. Natural autoantibodies are present from birth and mainly of the IgM isotype, less frequently of the IgA or IgG isotype [11,28]. They are encoded by unmutated V(D)J genes and have a moderate affinity to self-molecules. Natural autoantibodies are thought to play a key role in immune homeostasis. Polyreactive IgM autoantibodies are involved in early immune responses. Moreover, natural autoantibodies enhance phagocytic clearance of apoptotic cells and cell debris and, thus, prevent uncontrolled inflammation [29]. Furthermore, natural autoantibodies mask antigens and, as a result, prevent binding of pathologic autoantibodies which could promote autoimmunity. In addition, natural autoantibodies can suppress inflammatory responses to Toll-like receptor agonists [30]. In the peripheral immune system, binding of autoreactive T and B cells to endogenous antigens promotes both T- and B-cell survival. For further information on the function of natural autoantibodies, we refer to reviews published by Siloşi et al. [27], Silverman et al. [30] and Elkon and Casali [31]. Summarizing the described mechanisms, evidence suggests that natural autoantibodies might ameliorate risk and severity of autoimmune diseases [32,33,34] and may therefore have a therapeutic potential in autoimmune diseases.

Besides these natural autoantibodies, there are pathogenic somatically mutated autoantibodies—class-switched to IgG isotype—which are secreted by autoreactive B cells. Rarely, these pathogenic autoantibodies can also display the IgM or IgA isotype. These autoantibodies show a high affinity to their respective antigen and might be involved in autoimmune disease. In general, B cells secreting these autoantibodies can contribute to pathomechanisms in autoimmune diseases via autoantibody-dependent and autoantibody-independent pathways.

Autoantibody-independent mechanisms of B cells involve the secretion of proinflammatory cytokines [35], the formation of ectopic germinal centers in inflamed tissues [36,37] and the role of B cells as antigen-presenting cells. Especially via the latter mechanism, autoreactive B cells are involved in the pathophysiology of various autoimmune diseases. Pathologies in autoimmune diseases are often characterized as T-cell-mediated. However, this interpretation might underestimate the complex interactions of B and T cells and the role of B cells as antigen-presenting cells in CD4+ T-cell activation [38].

Autoantibody-dependent mechanisms involve complement activation and activation of neutrophils and NK cells by immune complexes composed of autoantibody and autoantigen. Activation of the classical complement pathway results in the release of C3a and C5a which promote the release of proinflammatory cytokines, migration of immune cells and upregulation of FcR on effector cells. The upregulation of FcR on effector cells augments the antibody-dependent cell-mediated cytotoxicity (ADCC). Autoantibodies facilitate antigen uptake by antigen-presenting cells such as monocytes and dendritic cells. By this mechanism, autoantibodies enhance T-cell responses to the respective antigens, which is of great importance in T-cell-mediated autoimmune diseases. Moreover, autoantibodies can stimulate and inhibit receptor function. Characteristics of natural and pathogenic autoantibodies are summarized in Table 1.

In the development of diagnostic methods for the detection of autoantibodies, it is crucial to differentiate natural autoantibodies and autoantibodies that mediate pathologies. Criteria for this distinction were first proposed in 1993 and further elaborated in subsequent years [39,40]. This is particularly important, as B-cell activation and hypergammaglobulinemia, which occur in the context of many chronic inflammatory diseases, could also increase the level of natural autoantibodies without inducing or maintaining pathologies. Recently, studies suggested the use of serum IgG levels to differentiate whether increased autoantibody levels are due to a non-specific B-cell activation or an antigen-specific autoimmune reaction [40,41].

### 2.3. Autoreactive B Cells and Autoantibodies

Several mechanisms have been described that may contribute to the maturation of autoreactive B cells and subsequent secretion of autoantibodies in SSc.

A physiological mechanism by which autoantibody formation occurs is the presence of polyreactive immunoglobulins that can bind diverse antigens. This reduced specificity can be advantageous in the immune defense, as a B cell can thus ward off different pathogens with similar antigens. However, a reduced specificity exhibits substantial cross-reactivity with endogenous antigens. Thus, polyreactive immunoglobulins are often autoreactive [42,43,44]. Along with this, the molecular mimicry hypothesis describes that T and B cells with specificity for an antigen of a pathogen also cross-react with self-antigens. This hypothesis was investigated in greater depth in SSc, as higher levels of antibodies to human cytomegalovirus proteins were detected in SSc than in healthy controls [45]. In addition, defined antigen-specific antibodies for human cytomegalovirus were found to be associated with autoantibody specificities of SSc [46,47]. Another interesting study investigating the molecular mimicry hypothesis was published in 2020 by Gourh et al. [48]. This study suggested a link between HLA alleles, peptides of viruses that infect amoebas or algae and anti-fibrillarin, anti-topoisomerase I and anti-centromere autoantibodies in African American and European American patients with SSc based on molecular mimicry.

A further mechanism that can predispose the formation of autoantibodies describes a loss of self-tolerance in the B-cell compartment, e.g., due to deficient negative selection or excessive stimulation, together with increased antigen expression or excessive antigen release due to cell damage. Exemplarily, the relationship between increased antigen expression and increased autoantibody formation could be shown for the AT1R. An increased expression of AT1R in peripheral blood mononuclear cells, skin and lungs corresponds to increased AT1R autoantibody levels in SSc [28].

Moreover, several aberrations promoting autoreactivity and autoantibody secretion were identified in SSc. Accordingly, Glauzy et al. demonstrated a deficiency in central and peripheral B-cell tolerance checkpoints in patients with SSc, promoting the development of autoreactive naïve B cells [49].

Moreover, SSc patients display higher levels of the B-cell survival factor BAFF and B cells of SSc patients exhibit increased levels of the BAFF receptor [50]. Complementarily, a genome-wide association study revealed an association between a variant in the BAFF gene (TNFS13B) and multiple sclerosis, as well as systemic lupus erythematosus. This variant is associated with increased levels of soluble BAFF, lymphocytes and immunoglobulins promoting humoral immunity [51].

Among others, these aberrations might contribute to the increased relative count of B cells and a disturbed composition of the B-cell compartment in SSc [52]. Several studies revealed distinct alterations of B-cell subsets that might promote autoreactivity. Firstly, SSc patients exhibit decreased levels of regulatory B cells and regulatory memory B cells [50,53]. This decrease is especially prominent in patients with pulmonary arterial hypertension [54]. Regulatory B cells secrete the anti-inflammatory cytokine IL-10 and they inhibit the induction of antigen-specific inflammatory reactions [52]. Accordingly, in SSc, levels of circulating regulatory B cells negatively correlated with anti-centromere and anti-topoisomerase I autoantibody levels as well as disease activity [50]. Moreover, regulatory B cells inhibit CD4+ Th1 and Th17 cell differentiation and cytokine secretion and induce regulatory T cells [55]. In addition, regulatory B cells are thought to participate in regulation of T_fh_ cells in SSc. Accordingly, levels of regulatory B cells and T_fh_ cells are negatively correlated [54].

Moreover, evidence indicates a decrease in plasmablasts and memory B cells due to an increased sensitivity to Fas-mediated apoptosis. Although the number of memory B cells (CD19+CD27+) decreases, these cells show an activated phenotype with increased expression of CD80 and CD86, which are co-stimulatory molecules of B cells [56]. Among memory B cells, distinct subsets are aberrated. Exemplarily, CD21low B cells were found to be increased in SSc patients, especially with visceral vascular manifestations, compared to healthy controls [57,58]. However, data on an increased prevalence of pulmonary arterial hypertension in patients with more than 10% of CD21low B cells are contradictory [54,58]. CD21low B cells are thought to have a high autoreactive potential as these cells express high levels of activation markers and act as antigen-presenting cells. An increase in the number of these cells has also already been described in other autoimmune diseases [59,60,61,62].

Probably, compensatory to the decrease in memory B cells, the number of naïve B cells (CD19+CD27-) is increased [56,63].

Furthermore, a change in the gene expression of B cells towards increased activity was detected [8]. Correspondingly, in SSc patients, expression of regulator molecules controlling B-cell responses are altered [64]. Increased CD19 expression was shown on naïve and memory B cells compared to healthy controls [56,65]. Experiments with CD19 transgenic mice showed that these mice produced elevated levels of autoantibodies, including SSc-specific autoantibodies, but without inducing a pathological phenotype [66,67,68,69].

To summarize, these aberrations in the B-cell compartment result in polyclonal B-cell hyperreactivity, hypergammaglobulinemia [70] and autoantibody production in SSc. However, in SSc, autoantibodies are detectable years before clinical disease manifestations [11]. Therefore, it is challenging to identify the mechanisms that lead to the loss of self-tolerance.

## 3. Which Autoantibodies Might Be Useful as Predictors of Disease Course and Which Role Do Autoantibodies Play in the Pathophysiology of SSc?

Because of the heterogeneity of SSc, biomarkers are essential for stratifying patients and predicting an individual disease course. Therefore, the field of biomarker research is an emerging field of investigation in SSc. Though the potential role of autoantibodies as biomarkers is frequently investigated in SSc, functional data on autoantibody-mediated pathomechanisms are rare and poorly understood. An improved understanding of autoantibody-mediated pathomechanisms is essential to identify appropriate targets for therapeutics that remove specific autoantibodies.

In the following section, we present clinical associations, roles as biomarkers and influences on pathomechanisms in SSc for autoantibodies against nuclear antigens (ANAs), anti-neutrophil cytoplasmic antibodies (ANCA), anti-phospholipid antibodies (aPL) and autoantibodies recognizing GPCRs.

### 3.1. Autoantibodies against Nuclear Antigens (ANAs)

A diagnostically important feature of SSc is the presence of circulating ANAs directed against nuclear or nucleolar proteins involved in transcription, splicing or cell proliferation. ANAs can be detected by indirect immunofluorescence. However, this technique is insufficient to identify specific ANAs except for anti-centromere antibodies. Therefore, additional techniques such as ELISA, immunodiffusion or Western immunoblotting can be used to determine the patients’ individual specific antigenic targets [71].

The determination of ANAs as well as specific ANAs is well-established in diagnosing SSc and is part of the ACR/EULAR 2013 classification criteria for SSc [72]. ANAs can be detected in >90% of SSc patients [73,74]. The group of ANA-negative patients constitutes a distinct clinical subtype characterized by predominantly male patients with a severe disease course, involvement of the lower gastrointestinal tract with corresponding symptoms and less vasculopathy [75]. Another study revealed an association between ANA negativity and gastric antral vascular ectasia (GAVE) [76].

Until now, a variety of SSc-specific ANAs (anti-topoisomerase I, anti-centromere, anti-RNA polymerase III, anti-Th/To, anti-eukaryotic initiation factor 2B (anti-eIF2B), anti-U11/U12 RNP) and ANAs not exclusively specific for SSc (anti-Pm/Scl-100, anti-Pm/Scl-75, anti-Ro52, anti-Ku, anti-fibrillarin (U3-RNP), anti-U1 RNP, anti-NOR90/hUBF, anti-RuvBL1/2) have been identified. Each of these autoantibodies has been associated with a unique set of disease manifestations, enabling the prediction of disease course, development of organ manifestations and an individual prognosis [77]. To further explain clinical differences of ANA subspecificities, Clark et al. performed a transcriptional and proteomic analysis of blood and skin of SSc patients with anti-topoisomerase I and anti-RNA polymerase III antibody specificities, revealing pathogenetic differences between ANA-based subgroups [78]. For the association of SSc-specific ANAs with corresponding clinical phenotypes, we refer to recently published reviews, e.g., by Cavazzana et al. [71] or Stochmal et al. [79]. In addition, recently, further ANAs were identified in subsets of SSc patients. These autoantibodies target telomerase and shelterin proteins and show an association with severe interstitial lung disease [80]. Moreover, detection of ANA subspecificities in patients suffering from SSc-associated diseases such as primary biliary cholangitis can be used to identify patients at increased risk for developing SSc [81,82].

A remarkable observation is that the ANA titers and specific ANAs remain relatively stable over the disease course, which makes them a valuable diagnostic tool [83].

Moreover, in most patients, only few ANA subspecificities are detected in parallel, so mutual exclusivity is assumed. Against this background, however, the simultaneous occurrence of various SSc-specific autoantibodies has been detected in several studies in small subgroups of patients [84,85]. In particular, the joint occurrence of anti-centromere and anti-topoisomerase I antibodies was investigated. Depending on the ethnicity of the patients examined and the techniques used to determine the ANA subspecificities, divergent results were found. So far, no clear clinical cluster has been identified that is associated with the simultaneous presence of anti-centromere and anti-topoisomerase I autoantibodies. Currently, there is no proven pathophysiological concept that explains the relative mutual exclusivity of the autoantibodies. Hypotheses suggest that the different ANA subspecificities might be epiphenomena based on different environmental conditions or due to differences in antigen processing by B cells showing associations with particular HLA alleles [86,87,88].

Currently, due to increasing possibilities for quantitative measurements of ANA subspecificities, it has been investigated whether these measurements could provide additional information to predict disease progression. These analyses revealed evidence that levels of anti-topoisomerase I antibodies are associated with disease severity [89,90]. However, the results of large multicenter studies need to be awaited.

In addition to the well-established role of ANAs in diagnostics and predicting disease progression, data on a pathogenic role in disease onset or maintenance are rare in SSc. In 1996, Rudnicka et al. demonstrated an altered activity of the topoisomerase I enzyme in SSc and suggested to evaluate topoisomerase I inhibitors as a potential therapeutic approach [91]. Moreover, studies suggest a pathogenic role of anti-topoisomerase I autoantibodies. This assumption is based on the observation that anti-topoisomerase I autoantibodies bind to the cellular surface of fibroblasts [92]. Moreover, binding of anti-topoisomerase I autoantibodies to fibroblasts stimulates adhesion and activation of monocytes in vitro [93]. Another study brought further evidence for a pathogenicity of anti-topoisomerase I and anti-centromere autoantibodies by stimulating human dermal fibroblasts with these autoantibodies and, as a result, inducing an increase in pro-fibrotic markers and apoptosis resistance [94].

Although these findings do not necessarily prove a substantial role of ANA in disease pathogenesis, these data provide first evidence for the use of immunosuppressive treatment in early SSc.

### 3.2. Anti-Neutrophil Cytoplasmic Antibodies (ANCAs)

The term ”anti-neutrophil cytoplasmic antibodies” (ANCAs) refers to autoantibodies against enzymes in primary granules of neutrophils or lysosomes of monocytes. ANCA subspecificities differ by indirect immunofluorescence, namely a cytoplasmic pattern (c-ANCA, e.g., autoantibodies against proteinase 3 (PR3)), a perinuclear pattern (p-ANCA, e.g., autoantibodies against myeloperoxidase (MPO)) and an atypical pattern showing aberrant patterns or a combination of c- and p-ANCA patterns (a-ANCA/x-ANCA). In addition to PR3 and MPO, ANCAs can be directed against further proteins such as elastase, cathepsin G, lactoferrin, α-enolase, catalase, azurocidin or actin [95]. The detection of ANCAs against PR3 and MPO is the hallmark of ANCA-associated vasculitis (AAV). In AAV, ANCAs lead to an activation of neutrophils primed by complement fragment C5a or cytokines (TNF-α, IL-1β), resulting in a translocation of PR3 or MPO to the cellular surface. After interaction between ANCAs with the target antigens PR3 or MPO and Fcγ receptors IIa or IIIb, intravascular near-wall degranulation of neutrophils and subsequent endothelial cell damage occurs [96,97,98]. ANCA-induced monocyte activation is mediated by a similar mechanism [99,100]. Subsequently, leukocyte migration and organ infiltration lead to secondary organ damage [101]. However, ANCAs can be detected in various other diseases (e.g., inflammatory bowel diseases, autoimmune hepatitis, primary sclerosing cholangitis), too [102].

Detection of ANCAs is a common phenomenon in SSc: the prevalence of ANCAs in SSc ranges up to 35% [103]. However, most studies report ANCA prevalence in 5–12% of SSc patients. The main antigenic targets are PR3 and MPO [104,105,106]. Although the presence of ANCAs is a common phenomenon in SSc, the development of systemic vasculitis in patients is rare [106,107]. The occurrence of systemic small-vessel vasculitis in SSc is grouped under the term SSc-AAV and is associated with a severe clinical phenotype involving the kidney, lung, peripheral and, rarely, the central nervous system, typically with a microscopic polyangiitis-like disease pattern [106]. In addition, patients may develop necrotizing vasculitis with critically reduced acral perfusion. Patients with renal manifestations typically present with pauci-immune glomerulonephritis or rapidly progressive glomerulonephritis. Patients with pulmonary involvement typically develop alveolar hemorrhage [106]. To date, there is limited evidence regarding promising treatment options for SSc-AAV. Randomized controlled trials are lacking. Possible treatment options that have been used so far include high-dose steroids, cyclophosphamide, rituximab, and plasma exchange. Before using high-dose steroids, the potential risk of developing a renal crisis should be considered in SSc [108]. In addition to autoantibodies against PR3 and MPO, autoantibodies with BPI and cathepsin G as antigenic targets have also been described in SSc [103]. Patients with ANCAs against BPI display lower skin scores.

### 3.3. Anti-Phospholipid Antibodies (aPL)

aPL comprise a group of antibodies directed against phospholipids and their cofactors, such as β2-glycoprotein I, prothrombin, annexin V and protein C or S. Phospholipids are components of the cell membrane and, together with the cofactors mentioned, play a role in hemostasis. In recent years, progress has been made in deciphering the pathomechanisms by which the binding of these autoantibodies to the protein–phospholipid complexes leads to increased blood coagulation, especially for the anti-β2-glycoprotein I antibodies.

aPL binding to β2GPI evoke a prothrombotic situation in endothelial cells through increased expression of adhesion molecules and tissue factor as well as reduced expression of the tissue factor pathway inhibitor [109,110]. In addition, complement activation is described [111]. Furthermore, in platelets, incubation with antibodies results in an activation of the glycoprotein IIa/IIIb (GPIIb/IIIa) receptor [112]. Moreover, activation of neutrophil granulocytes and monocytes by aPL has also been demonstrated [113,114,115].

aPL can be detected primarily without association to an underlying disease. In addition, aPL may be secondarily associated with autoimmune diseases such as systemic lupus erythematosus, rheumatoid arthritis, infections and cancers, but also with the use of medications such as oral contraceptives.

The detection of aPL also occurs frequently in SSc, with a prevalence of 9–20% [116]. In SSc, an association of anti-β2 glycoprotein 1 (β2GP1) positivity with active digital ulcerations has been demonstrated [117]. Moreover, a meta-analysis by Merashli et al. revealed an association between antibody positivity and pulmonary arterial hypertension, renal disease, thrombosis, miscarriage and digital ischemia compared to patients without aPL [118]. The association to venous thrombosis and miscarriage was confirmed in a meta-analysis by Sobanski et al. [116].

### 3.4. Autoantibodies Recognizing G-Protein-Coupled Receptors, Growth Factors and Their Respective Receptors

In addition to ANAs, ANCAs and aPL described in the previous sections, a new group of autoantibodies has recently been described which, in addition to their significance as biomarkers, are also becoming increasingly important in the pathophysiology of SSc. These comprise self-reactive autoantibodies recognizing GPCRs, growth factors and their respective receptors [26]. Anti-GPCR autoantibodies are present in healthy individuals and are thought to play a role in the regulation of immune cell homeostasis. These autoantibodies form a distinct, disease-specific network. It is hypothesized that this network reflects the patient’s individual exposure to a specific environmental condition [28].

To investigate functional effects of the autoantibody network in a disease, several in vitro and in vivo technologies were established (Figure 2). In vitro stimulation of cell lines or isolated cells with purified IgG fractions of SSc patients and healthy donors is an established technology which led to the formation of autoantibody classes with the same cellular target, e.g., anti-endothelial cell autoantibodies (AECA) or anti-fibroblast autoantibodies (AFA). Since it has not yet been possible to purify target-specific autoantibodies, the effects of target-specific autoantibodies can only be studied by using receptor blockers. Applying this technology, Murthy et al. stimulated cells of the monocytic cell line THP-1 with IgG purified from patients with SSc. Stimulation with SSc–IgG induced a change to a pro-fibrotic and pro-inflammatory phenotype with IgG donor–individual alterations. Moreover, SSc–IgG induced pathways including AP-1, TAK/IKK-β/NFκB and ERK1/2, driving secretion of CCL18 and CXCL8 from stimulated cells [119].

Moreover, several strategies have been established to transfer autoantibody-induced pathologies to animal models and build a platform to further investigate the respective pathomechanisms. These strategies involve the transfer of (a) serum [120], (b) IgG purified from serum [121] and (c) PBMC from diseased patients and healthy donors. A further strategy involves immunization with special agents, e.g., GPCR-overexpressing membrane extracts, (d), leading to the generation of autoantibodies and secondary to pathogenic effects [122,123]. These strategies were also applied to develop animal models of SSc mirroring autoantibody-mediated pathologies.

In 2014, Becker et al. transferred IgG purified from serum of SSc patients to C57BL/6J mice. These mice developed histological signs of an inflammatory pulmonary vasculopathy [124]. In addition, SSc–IgG positive for anti-AT1R and anti-ETAR autoantibodies induced increased neutrophil counts in bronchoalveolar fluid, pulmonary cellular infiltrations and cellular density after repeated passive transfer to C57BL/6J mice [125]. In a further study, a monoclonal AT1R autoantibody was injected into mice, which led to skin and lung inflammation. Interestingly, skin and lung inflammation did not occur in mice deficient in AT1Ra/b. This observation suggests a compelling involvement of autoantibody–receptor interaction in pathophysiological mechanisms [126].

In 2021, Yue et al. transferred PBMC of patients with SSc and granulomatosis with polyangiitis (GPA), as well as that of healthy donors, to Rag2^-/-^/IL2rg^-/-^ mice. Subsequently, mice engrafted with PBMC developed an ANA pattern similar to the respective donor [127]. In a subsequent study, performed by the same group, membrane-embedded human AT1R or empty membranes as controls were transferred to C57BL/6J mice [126]. Immunization with membrane-embedded human AT1R resulted in detectable levels of AT1R autoantibodies in mice and induced skin and lung inflammation as well as skin fibrosis.

These studies support the pathophysiological concepts of SSc described in the previous sections. Moreover, these studies provide animal models of SSc resembling human pathophysiology and, thus, these animal models enable evaluation of potential therapeutic approaches.

#### 3.4.1. Functional Autoantibodies against GPCR

In the following, we summarize data on specific autoantibodies targeting GPCR. GPCRs form a large family of receptors in vertebrates and are characterized by seven transmembrane domains with intervening extracellular and intracellular amino acids. These receptors are named for their interaction with G-proteins, which mediate intracellular signal transduction. In addition to the G-proteins, G-protein-independent pathways for signal transduction have been described. In recent years, an increasing number of GPCRs that are recognized by autoantibodies have been identified. Studies have shown that the occurrence of these autoantibodies is not linked to diseases, but that physiological levels of autoantibodies that recognize GPCRs can also be detected in healthy controls [26]. Subsequently, altered levels of these autoantibodies were described for numerous diseases, e.g., solid organ or stem-cell transplantations, cardiovascular diseases, cancer, neurological, endocrine, pulmonary or rheumatic systemic diseases.

##### Anti-AT1R and Anti-ETAR Autoantibodies

Anti-AT1R Autoantibodies are well-investigated autoantibodies involved in the pathophysiology of several diseases, e.g., kidney transplant rejection, preeclampsia, diabetes mellitus [128], lupus nephritis [129] or COVID-19 disease [130]. Autoantibodies directed against AT1R were first described in 1999. Specifically, Wallukat et al. described the occurrence of autoantibodies that recognize AT1R in patients with preeclampsia [131]. AT1R is a GPCR whose binding of the endogenous ligand angiotensin results in Gq-mediated calcium release, β-arrestin-mediated cell signaling and the production of reactive oxygen species. Further research in 2005 demonstrated the role of these antibodies as biomarkers predicting rejection in kidney transplantation [132]. In the meantime, the measurement of AT1R autoantibodies in transplantation medicine is well-established in clinical routine. Furthermore, in addition to the role of AT1R autoantibodies as biomarkers, the pathophysiological mechanisms mediated by the antibodies in transplant rejection could be deciphered. Anti-AT1R autoantibodies bind to two different epitopes, namely AFHYESQ and ENTNIT, and act as allosteric agonists at the AT1R. Thus, they lead to sustained activation [132,133,134]. Interaction of the anti-AT1R autoantibody with the receptor results in vasoconstriction as well as the formation of proinflammatory, profibrotic and procoagulatory conditions in the microvascular circulation. The proinflammatory and procoagulatory processes are activated by the receptor activation-induced phosphorylation of extracellular signal-regulated kinase 1/2 (ERK1/2) and subsequent activation of the transcription factors activator protein 1 (AP-1) and nuclear factor kB (NFκB) in the endothelium and smooth muscle vascular cells. Expression of AT1R on the surface of immune cells, especially polymorphonuclear leukocytes, monocytes, T and B lymphocytes, results in pro-inflammatory gene expression (IL-1, IL-6, IL-8, IL-17, TNF-α, IFN-γ), which may even maintain AT1R expression on endothelial cells [135,136,137]. Moreover, microvascular inflammation and vasoconstriction induced by anti-AT1R autoantibodies promotes thrombosis, endarteritis and fibrinoid necrosis. Here, the interaction between receptor and autoantibody depends on the AT1R expression level. The AT1R expression level varies due to genetic and non-genetic factors. Genetic factors involve polymorphisms in the AT1R gene [138]. Non-genetic factors that increase AT1R expression are inflammation, endothelial damage or disturbances in microcirculation [136,139]. For further information on anti-AT1R autoantibodies in kidney transplant rejection, we refer to a review published by Sorohan et al. [140].

Interestingly, similar mechanisms are assumed to be involved in the pathogenesis of preeclampsia. Here, the epitope, targeted by anti-AT1R autoantibody, is AFHYESQ which has also been described in kidney transplant rejection [131]. A further mechanism proposed for mediating effects of anti-AT1R autoantibodies in preeclampsia is the long-term presence of anti-AT1R autoantibodies which reduces aldosterone production in vitro [141].

The similarity of the autoantibody binding sites and the secondary processes in kidney transplant rejection and preeclampsia suggest similar pathophysiologic processes in other diseases. However, the epitope to which anti-AT1R autoantibodies—which mediate pathologies—bind in SSc has not yet been identified.

In SSc, elevated levels of angiotensin II (AngII) and endothelin-1 (ET-1) were detected in blood and tissue samples [142,143]. Therefore, the corresponding receptors were discussed as potential therapeutic targets; endothelin-1 receptor blockers are recommended for treatment of pulmonary arterial hypertension [144]. Moreover, bosentan showed beneficial effects in the treatment of Raynaud’s phenomenon and SSc-related digital ulcers [145,146,147,148].

In the majority of SSc patients, anti-AT1R and anti-ETAR autoantibodies are detectable. In addition, levels of anti-AT1R and anti-ETAR autoantibodies show a strong correlation with each other. Furthermore, both autoantibodies display the ability to exert functional activity at the respective receptor [28]. Regarding functional activity, in vitro experiments with IgG and receptor blockers for AT1R and ETAR proved that anti-AT1R and anti-ETAR autoantibodies bind to endothelial cells, exert phosphorylation of extracellular signal-regulated kinase 1/2 and increase TGFβ gene expression [149]. Moreover, Kill et al. demonstrated that anti-AT1R and anti-ETAR autoantibodies activate human microvascular endothelial cells in vitro, promoting secretion of proinflammatory chemokines and increased expression of adhesion molecules, enabling migration of neutrophils through an endothelial cell layer. Furthermore, using the same experimental setup, Kill et al. induced profibrotic processes in fibroblasts [125]. Günther et al. investigated the effects of PBMC stimulation with IgG from SSc patients and healthy controls, revealing an increased induction of IL-8 and CCL18 by SSc–IgG compared to healthy controls. These effects could be diminished by adding AT1R and ETAR blockers to the experimental setup [150].

In addition, anti-AT1R and anti-ETAR autoantibodies amplified vasoconstrictive effects of Ang II and ET-1 in small-vessel myography of intralobar pulmonary rat artery ring segments [124]. The amplification of Ang II- and ET-1-mediated effects by anti-AT1R and anti-ETAR autoantibodies was confirmed in in vitro analyses using the technology ”dynamic mass redistribution” [126].

High levels of these autoantibodies showed an association with severe disease complications. Anti-ETAR autoantibodies—together with acute digital ulcers or ulcers in a patient’s history—can be used to predict development of subsequent digital ulcers [151]. Moreover, anti-AT1R and anti-ETAR autoantibodies can predict the development of pulmonary arterial hypertension in SSc. Moreover, comparing levels of anti-AT1R and anti-ETAR autoantibodies in forms of pulmonary hypertension revealed highest levels for pulmonary arterial hypertension in SSc and connective tissue diseases. The same study revealed anti-AT1R and anti-ETAR autoantibodies as predictors of mortality in SSc [124]. Further information on the role of anti-AT1R and anti-ETAR autoantibodies are summarized in a review by Cabral-Marques and Riemekasten [152].

##### Anti-Muscarinic-3 Acetylcholine Receptor (M3R) Autoantibodies

Anti-M3R autoantibodies have been associated with intestinal dysmotility: a cardinal pathological condition and cause of the most gastrointestinal manifestations in patients with SSc [153]. Kumar et al. suggested a model of sequentially developing dysmotility: anti-M3R autoantibodies initially inhibit the release of acetylcholine (Ach) at the myenteric cholinergic neurons, inducing neuropathy through the blockage of cholinergic neurotransmission. Consequently, myopathy develops as a result of an anti-M3R autoantibody-related inhibition of Ach at the smooth muscle cells of the gastrointestinal tract. Smooth muscle fibrosis and atrophy ensue [154]. It can thus be hypothesized that an early and sustained elimination of anti-M3R autoantibodies could possibly lead to a reversal of SSc-associated dysmotility at the neuropathic and myopathic stages. Indeed, Kumar et al. presented evidence that application of IVIGs decreases binding intensity of anti-M3R autoantibodies and, thus, could probably decrease SSc-related gastrointestinal symptoms [154]. Accordingly, Raja et al. confirmed a significant improvement of gastrointestinal symptoms after repeated IVIG administration [155].

##### Anti-CXCR3 and Anti-CXCR4 Autoantibodies

Weigold et al. showed that autoantibody levels differ among subgroups of patients suffering from SSc, with dcSSc patients having the highest levels of autoantibodies directed to the N-terminal domain of CXCR3 and CXCR4. Comparable to anti-AT1R and anti-ETAR autoantibodies, anti-CXCR3 and anti-CXCR4 autoantibody levels also correlate with one another. Moreover, in SSc patients with interstitial lung disease, levels of autoantibodies directed to the N-terminus of CXCR3 and CXCR4 were lower in patients with progressive disease than in patients with stable disease [156]. Therefore, these autoantibodies might be a valuable tool to predict disease course of SSc patients with interstitial lung disease. A proposed hypothesis for an association between low autoantibody levels and disease progression is that the corresponding autoantibodies might be predominantly present in the tissues and, accordingly, detection of autoantibody levels in the blood may give lower levels [28]. So far, however, no studies have been conducted to bring evidence to this hypothesis. Currently, it is unclear whether autoantibodies directed to CXCR3 and CXCR4 exhibit functional activity. As shown for the AT1R autoantibody, functional activity of an autoantibody depends on the respective epitope. Therefore, Recke et al. applied a peptide-based epitope mapping for CXCR3. In this analysis, they could show differences in epitopes of anti-CXCR3 autoantibodies between SSc and healthy controls. Whereas autoantibodies from SSc patients preferentially bind to intracellular CXCR3 epitopes, autoantibodies of healthy controls bind to epitopes in the N-terminal domain [157].

##### Anti-PAR-1 Autoantibodies

Further potentially interesting autoantibodies in SSc are anti-PAR-1 autoantibodies. A first hint that anti-PAR-1 autoantibodies might exhibit functional activity in SSc was based on the observation that IL-6 release of HMECs stimulated with IgG from SSc patients could be reduced by a PAR-1 inhibitor. Moreover, stimulation of HMECs with IgG from SSc patients resulted in an increased expression of phosphorylated pAKT, p70S6K and pERK1/2, whereas this increase in expression was not observed after stimulation with IgG from healthy controls. Further experiments revealed an increased transcriptional activity of c-FOS and AP-1 after stimulation with SSc–IgG, which finally results in IL-6 mRNA expression and, subsequently, secretion of IL-6. To sum this up, anti-PAR-1 autoantibodies resemble the signaling pathway of thrombin, one of the natural ligands, which also leads to IL-6 secretion after binding to PAR-1 [158]. However, currently, the role of IL-6 in scleroderma renal crisis is obscure. Additional evidence for the pathomechanism described here provides a study showing an improvement in creatinine levels during therapy with tocilizumab in scleroderma renal crisis [159].

## 4. Which Therapeutic Approaches Have Been Evaluated to Target Autoantibody-Mediated Pathologies in SSc?

Various attempts have been conducted to prevent the pathogenic effects mediated by autoantibodies in diseases. As autoantibodies represent a cross-disease pathomechanism, therapeutic approaches developed in the field of other diseases might be transferrable to the treatment of SSc. In the following, we list possible therapeutic approaches that have already been investigated in SSc or could be applied to SSc in the future (Figure 3).

### 4.1. B-Cell- and Plasma Cell-Mediated Strategies

As described, B cells play a critical role also in autoimmune diseases that are traditionally viewed as T-cell-mediated. Therefore, B-cell-targeting drugs are an emerging research area leading to the development of different therapeutic strategies. These include the elimination of defined cell subsets of the B-cell compartment, the neutralization of B-cell survival factors (e.g., BAFF and APRIL) and the prevention of the formation of ectopic germinal centers using antibodies against the lymphotoxin-β receptor.

#### 4.1.1. Anti-CD19 Antibody

Cell subsets of the B-cell compartment express different surface proteins, so that a depletion of defined B-cell subsets can be achieved by targeting a specific protein. A potential target applying this mechanism of action is CD19, a B-cell surface antigen expressed from pre-B cells through plasmablasts and in some plasma cells. CD19 is targeted by inebilizumab/MEDI-551, a humanized monoclonal antibody that mediates ADCC. The application of MEDI-551 was investigated in a phase I multicenter, randomized, double-blind, placebo-controlled, single escalating dose study in SSc. One of the 24 study participants receiving treatment with MEDI-551 died during the course of the study from a cause which was not attributed to the drug. In general, MEDI-551 displayed a tolerable safety profile and achieved a dose-dependent depletion of circulating B cells and plasma cells. Assessments of the mRSS suggest a beneficial therapeutic effect of MEDI-551 on skin fibrosis [160].

#### 4.1.2. Anti-CD20 Antibody

One of the best-studied B-cell-targeting therapeutics in SSc is rituximab. Rituximab is a chimeric anti-CD20 antibody. CD20 is expressed from cells of the pre-B cell stage to the pre-plasma cell stage. Thus, rituximab achieves a long-lasting, almost complete B-cell depletion in the blood and tissues. Six months after therapy initiation with rituximab, only a small number of CD19+ cells are detectable, which are predominantly IgA plasmablasts of mucosal origin [161]. Therapeutic efficacy of rituximab was assessed for several disease manifestations in SSc. Recently, Zamanian et al. investigated efficacy for the treatment of pulmonary arterial hypertension in SSc compared to placebo. Though the study failed to reach the primary endpoint, namely improvement in 6 min walk distance (6MWD) 24 weeks after treatment initiation, patients treated with rituximab had a significantly improved 6MWD after 48 weeks of treatment (*n* = 58, [162]). In addition, the therapeutic potential of rituximab on SSc-associated ILD was assessed in two small randomized controlled trials versus placebo (*n* = 16, [163]) and versus cyclophosphamide (*n* = 60, [164]) and several small non-controlled retrospective and prospective trials. The results of these two randomized controlled trials and 18 further studies or conference abstracts on rituximab in SSc-associated ILD were analyzed in a meta-analysis in 2021 by Goswami et al. (*n* = 575, [165]). This meta-analysis proves a significant improvement of FVC and DLCO under treatment with rituximab. Moreover, patients treated with rituximab exhibited less infectious complications than control patients. A key adverse event during treatment with rituximab remains a high potential for allergic reactions. Therefore, second-generation anti-CD20 antibodies (e.g., ocrelizumab, obinutuzumab, veltuzumab and ofatumumab) have been developed. These second-generation anti-CD20 antibodies are humanized or even fully human and have a higher therapeutic potential compared to rituximab in vitro [166]. Currently, none of these second-generation anti-CD20 antibodies have been evaluated in SSc.

A major obstacle in therapy with rituximab is the persistence of autoreactive long-lived plasma cells, which can produce autoantibodies and, thus, sustain the autoimmune disease. In addition, Mahévas et al. showed that B-cell depletion promotes differentiation from short-lived to long-lived autoimmune plasma cells by altering the splenic milieu [167].

#### 4.1.3. Anti-BAFF Antibody

Rituximab treatment triggers the secretion of B-cell-activating factor (BAFF), which perpetuates autoreactive B cells in systemic lupus erythematosus [168]. Therefore, a possibility to prevent the persistence of long-lived plasma cells could be a combination therapy with a monoclonal anti-BAFF antibody (Belimumab). An interesting effect of therapy with BAFF inhibitors is that these depleted B effector cells secreting IL-6, but did not lead to a depletion of regulatory B cells [169]. Belimumab is already approved for the treatment of systemic lupus erythematosus. A phase II study investigating the effects of belimumab and mycophenolic acid versus placebo and mycophenolic acid was conducted in 20 patients with dcSSc. Patients who received belimumab did not develop significant improvement in skin thickness compared to the placebo group. Patients who responded to treatment with belimumab showed a decrease in the expression of profibrotic genes [170]. A phase II trial investigating the effects of a combination therapy of rituximab and belimumab is registered (NCT03844061).

#### 4.1.4. Proteasome Inhibitor

An alternative therapeutic strategy that leads to a depletion of plasma cells is the use of proteasome inhibitors (e.g., bortezomib). Proteasome inhibitors are approved for the treatment of multiple myeloma. Early studies evaluating the use of proteasome inhibitors in mouse models of SSc were based on the premise of antifibrotic properties. This assumption was based on in vitro experiments on human fibroblasts, which showed a decrease in collagen production and an increase in collagen degradation when proteasome inhibitors were applied [171]. In a study conducted on the effects of proteasome inhibitors in mouse models of SSc, there was no effect on lung inflammation, lung fibrosis or skin fibrosis [172]. A phase II study was conducted comparing the effects on skin fibrosis and forced vital capacity in lung function as well as the tolerability of bortezomib and mycophenolate versus placebo with mycophenolate. The results were better for the combination therapy of bortezomib and mycophenolate (NCT02370693). In addition, a study is currently underway to investigate the safety and tolerability of oral ixazomib in scleroderma-related lung disease patients. This study is currently in recruitment status (NCT04837131). When applying proteasome inhibitors, however, it must be taken into account that depletion of the plasma cell compartment can lead to serious side effects.

#### 4.1.5. Anti-CD38 Antibody

In 2019, Benfaremo et al. proposed an alternative treatment trial with an anti-CD38 antibody (e.g., daratumumab) for the treatment of SSc. CD38 is expressed on plasmablasts and plasma cells and might be useful in depletion of antibody-producing cells [173].

#### 4.1.6. Inhibitor of Bruton Tyrosine Kinase

Ibrutinib is an irreversible inhibitor of Bruton tyrosine kinase, which is involved in intracellular signaling in B lymphocytes. Ibrutinib prevents signaling through the BCR, which promotes cell apoptosis and disrupts B-cell adhesion and B-cell migration. In addition, ibrutinib downregulates the expression of CD20 by targeting the CXCR4/SDF1 axis [174]. Ibrutinib is used to treat B-cell malignancies. In SSc, one study investigated possible effects of ibrutinib on B-cell pathologies in vitro. In this study, ibrutinib reduced the production of IL-6 and TNF-α by B effector cells. With the application of only small doses of ibrutinib, the function of regulatory B cells could be preserved [175].

#### 4.1.7. Therapeutic Approaches Targeting PAMP- and DAMP-Mediated Activation of the B-Cell Compartment

Although the role of B-cell activation and the corresponding effects on antibody secretion via TLRs in the development and maintenance of SSc are poorly understood, this mechanism may also represent an interesting therapeutic strategy. In particular, this therapeutic strategy might be interesting because PAMP and DAMP contribute to the pathogenesis of SSc via further mechanisms. There are several therapeutic strategies that target PAMP- and DAMP-mediated activation of the B-cell compartment. These can either interfere with the interaction between ligand and TLR or with downstream signaling pathways. A summary of possible therapeutic strategies in SSc was provided by O’Reilly in 2018 [23].

### 4.2. Autologous Hematopoietic Stem-Cell Transplantation (aHSCT)

aHSCT is thought to eliminate autoreactive T and B cells by high-dose immunosuppression. Afterwards, reinfusion of autologous hematopoietic stem cells promotes reconstitution of a naïve, self-tolerant immune system. Thereby, aHSCT alters adaptive and innate immune systems [176]. Regarding the B-cell compartment, aHSCT induces a shift from memory B cells to naïve B cells and an increase in frequency of regulatory B cells [177,178]. Moreover, measurements of anti-topoisomerase I autoantibody levels after aHSCT reveal a decline [83].

aHSCT is recommended as a treatment option for patients with severe and rapidly progressive SSc refractory to immunosuppressive therapy. The first aHSCT was performed in 1997 in SSc [179]. Since then, around 500 aHSCTs have been reported [180] and more than 1000 SSc patients have been transplanted worldwide [181]. Furthermore, three randomized controlled trials have shown superiority of aHSCT versus intravenous cyclophosphamide therapy (ASSIST trial [182], ASTIS trial [183], SCOT trial [184]). In addition, a further non-interventional study (NISSC) confirmed the therapeutic efficacy of aHSCT [185]. SSc patients undergoing aHSCT yield significant improvements in survival, quality of life, skin fibrosis and lung function. However, patients undergoing aHSCT have an increased risk of infectious complications, especially for CMV reactivations and mycotic infections. A marker for the development of infectious complications is a lower number of B cells before aHSCT [186]. Further adverse events include the risk of engraftment syndrome and secondary autoimmune disorders [187]. Currently, different regimens have been compared to outweigh therapeutic efficacy and adverse events. So far, specific recommendations regarding transplant procedures are missing [181]. Recently, the German Society for Rheumatology (DGRh) suggested criteria for patient selection [188]. Moreover, optimal timing of aHSCT in SSc is still under discussion [180]. Therefore, timing of aHSCT is currently investigated in the UPSIDE study (NCT04464434).

### 4.3. Unspecific Approaches for the Removal of Antibodies

#### 4.3.1. Therapeutic Plasma Exchange, Plasmapheresis and Rheopheresis

The term ”therapeutic plasma exchange”, also called ”therapeutic apharesis”, describes a procedure where the patient’s blood is filtered and replaced, e.g., by albumin or fresh frozen plasma. Plasmapheresis is a related procedure removing less than 15% of blood volume and, thus, does not require fluid replacement. These technologies remove autoantibodies, immune complexes, cytokines or adhesion molecules from the blood [189]. Therapeutic plasma exchange removes approximately 65% of potential circulating pathogenic factors [190]. In SSc, therapeutic plasma exchange or plasmapheresis has been reported in more than 500 patients [189]. Evaluation of these technologies was conducted as case studies, small observational studies or in one of three prospective randomized clinical trials. Studies present mainly a favorable therapeutic effect on skin fibrosis, musculoskeletal symptoms, Raynaud’s phenomenon, healing of digital ulcers and organ manifestation.

For the treatment of SSc, plasmapheresis is often combined with further therapeutics such as ACE inhibitors or immunosuppressive agents, e.g., with prednisolone alone or in a triple therapy with oral [191] or intravenous cyclophosphamide [192]. Further alternatives are combination therapies with IVIGs. This combination of therapies makes it difficult to compare study results and to estimate the therapeutic effects that can be achieved through plasma exchange or plasmapheresis.

Newer protocols suggest rheopheresis, a double-filtration plasmapheresis, which does not require replacement of fluid and, thus, reduces the risk of anaphylaxis. Rheopheresis is usually applied in conditions with microcirculatory alterations because of its beneficial effect on blood and plasma viscosity, erythrocyte deformability and aggregation. Therefore, rheopheresis was thought to have a beneficial effect on Raynaud’s phenomenon and digital ulcers. However, data regarding effects of rheopheresis on microcirculation in SSc are contradictory [193,194].

Adverse events as a result of applying therapeutic plasma exchange, plasmapheresis or rheopheresis are rare [195,196]. Main adverse events to consider are complications due to venous access, hypocalcemia and hypovolemia. Excluding SSc, plasmapheresis reduced autoantibodies in serum and also in cerebrospinal fluid in several diseases [197].

#### 4.3.2. Immunoadsorption

An alternative therapeutic procedure for the removal of autoantibodies from a patient’s blood is a technology called immunoadsorption. In immunoadsorption, plasma and blood cells are separated in an extracorporeal circuit. The plasma is then passed through a high-affinity column. This procedure enables almost complete removal of human immunoglobulins and immune complexes from the patient’s blood. Immunoadsorption is thus more effective than plasma exchange or plasma pheresis. However, this can also lead to a greater decrease in non-pathogenic immunoglobulins. So far, there are no studies that have investigated immunoadsorption in SSc. Evidence for the application of immunoadsorption in connective tissue diseases was reviewed by Hohenstein et al. [198].

#### 4.3.3. Intravenous Gammaglobulin (IVIg)

IVIg is a blood product prepared from the serum of a great number of donors. Although the mechanism of action is not fully understood, a “high-dose” administration of 2 g/kg/month appears to have immunomodulatory effects, which is why the use of IVIGs in a number of autoimmune-mediated diseases has become widespread in recent decades [199,200]. The immunomodulatory mechanisms of action include inhibition of T-cell proliferation, modulation of apoptosis, inhibition of superantigen-mediated activation of T cells, inactivation of inflammatory factors such as TNF-a and IL-1α, effect on cytokine levels, inhibition of phagocytosis, enhancement of the catabolism of autoantibodies, effect on glucocorticoid receptor-binding affinity and inhibition of deposition of classical pathway components [119,201]. In SSc, in addition to the reduction of gastrointestinal symptoms, immunoglobulins can also help in decreasing skin thickness and reduce arthromyalgia and muscle weakness [202,203].

### 4.4. Specific Approaches for the Removal of Antibodies

#### 4.4.1. Selective Removal of Autoantibodies by Lysosomal Degradation

Several mechanisms are involved in the regulation of antibody concentrations in the human body. An interesting mechanism involves the neonatal Fc receptor (FcRn) which has a structure similar to the major histocompatibility complex (MHC) class I and also associates with β2-microglobulin. The FcRn is involved in the transport of maternal IgG to the fetus. This receptor regulates the transport of not only antibodies, but also other serum proteins such as albumin, within and across cells, by binding IgG at pH < 6.5 in early endosomes and releasing IgG at neutral pH, enabling IgG recycling. As a result, FcRn reduces lysosomal IgG degradation and, thus, modulates IgG concentrations in serum and throughout the body. Whereas engineering of the variable regions of IgG is a widely used approach to develop therapeutically effective antibodies, modulation of the Fc region of IgG represents an emerging research area. E.g., the Fc part of tixagevimab and cilgavimab (Evusheld, formerly known as AZD7442), which is used for prophylaxis of developing symptomatic COVID-19 disease in patients with increased risk of insufficient response to vaccination, was mutated to extend half-life [204]. Moreover, several drugs were fused to Fc portions to increase half-lives through FcRn-mediated recycling, e.g., etanercept (Enbrel) or abatacept (Orencia) [205]. Regarding autoantibodies, new therapeutic strategies focus on the blocking of FcRn–IgG interaction to reduce autoantibody concentrations. Summarized under the term ”Abdeg” technology, Fc fragments and antibodies were engineered to inhibit FcRn–IgG interaction [206]. In 2021, the first Fc-based inhibitor, namely efgartigimod (Vyvgart), was approved for treatment of generalized myasthenia gravis by the USA Food and Drug Administration (FDA). Moreover, this technology was evaluated in further diseases, e.g., primary immune thrombocytopenia or pemphigus vulgaris and foliaceus [207,208,209,210,211].

Another emerging therapeutic technology is summarized by the term ”Seldeg”, an abbreviation for ”selective degradation”. Seldegs enable the selective degradation of antigen-specific antibodies by binding, on one hand, to cell surface molecules via the targeting component and, on the other hand, to antigen-specific antibodies by the antigen component. After binding both components, the complex of the antigen-specific antibody and Seldeg is internalized and degraded in lysosomes. In animal models of several diseases, Seldegs were investigated. In an animal model of antibody-mediated exacerbation of experimental autoimmune encephalomyelitis, Seldegs binding to exposed phosphatidylserine or to FcRn were compared. Both Seldegs led to an amelioration of disease severity [212]. A key advantage of Seldegs compared to FcRn inhibitors is that Seldegs do not reduce antibody levels that are not target specific. Therefore, immunosuppressive effects of current treatments were not observed.

#### 4.4.2. Selective Removal of Autoantibodies Using Aptamer BC007

Aptamers are artificially created, short, single-stranded DNA or RNA oligonucleotides that are capable of binding to a specific molecule with a high affinity [213]. The ability to digitally share their sequence information and produce them using a template enables fast and economical manufacturing [214]. Their heat resistance provides an additional advantage in processing, transporting and sterilizing the aptamers [215]. Aptamers can not only be useful diagnostic tools for recognizing pathogens and cancer or for tracking environmental contamination, but they also exhibit therapeutic uses [216]. Pegaptanib sodium (Macugen; Eyetech Pharma/Pfizer), a drug against age-related macular degeneration, is an RNA aptamer against vascular endothelial growth factor 165 (VEGF-165) and the only so-far-approved aptamer drug [217]. However, many other aptamers are currently being studied and their application could revolutionize the treatment of many diseases. BC007, which was originally developed as a thrombin inhibitor [218], is the only known aptamer that is able to neutralize functional antibodies against GPCRs. Haberland et al. managed to neutralize antibodies against GPCRs that had been linked to cardiovascular pathologies such as dilated cardiomyopathy (DCM) and Chagas’ cardiomyopathy in vivo through the application of BC007 [219]. Wallukat et al. demonstrated the neutralizing efficiency of BC007 in serum of patients with DCM, where this serum was treated with the aptamer ex vivo. Moreover, antibody neutralization was achieved in vivo in spontaneously hypertensive rats [220]. Antibodies against GPCRs have also recently been associated with COVID-19 severity [221]. Hohberger et al. published a case report of a patient with long-COVID syndrome and positivity for antibodies against GPCRs to whom BC007 was applied intravenously. After a single application and during the subsequent four-week observation, an inactivation of GPCRs and a sustained improvement of fatigue, taste and retinal capillary microcirculation was detected [222]. Moreover, in fibrosis, a key feature of SSc, aptamers targeting downstream TGF-β signaling have been investigated [223]. These aptamers could also be a potential therapeutic approach for the treatment of SSc.

### 4.5. Therapeutics Targeting B-Cell-Secreted Cytokines

Tocilizumab is a humanized monoclonal antibody inhibiting the binding of IL-6 to the membrane and soluble IL-6 receptor. Therapeutic efficacy of tocilizumab has been investigated in a phase II (faSScinated study, *n* = 87 [224]) and a phase III study (focuSSced study, *n* = 212 [225]). Both studies did not reach their primary endpoint: mRSS improved under therapy with tocilizumab without reaching statistical significance compared to placebo. However, in both studies, tocilizumab preserved lung functionality with a significantly smaller decline in FVC than with placebo. Patients with short disease duration and elevated inflammatory markers responded to therapy with tocilizumab. The most common adverse events under therapy with tocilizumab in both studies were infections. Based on these results, the FDA approved tocilizumab for treatment of SSc-associated ILD in March 2021. Though both trials could not demonstrate significant improvement in mRSS compared to placebo, in vitro treatment of cultured skin fibroblasts from SSc patients with tocilizumab revealed a normalization of the genetic profile resulting in an inactive molecular and functional fibroblast phenotype [226]. For further information on the role of IL-6 in the pathophysiology of SSc and the potential therapeutic efficacy of the IL-6 inhibitor tocilizumab in further studies, we refer to a recently published review by Cardoneanu et al. [227].

## 5. Conclusions

So far, the mechanisms leading to the development of autoreactivity in the B-cell compartment and, secondarily, to the formation of autoantibodies are insufficiently understood. Similarly, investigations regarding the mechanisms by which autoantibodies functionally intervene in the pathology of SSc are still in their infancy. We expect that research on autoreactive B cells and autoantibodies will identify further autoantibodies that might serve as biomarkers. In addition, the understanding of autoantibody-mediated pathologies will grow. As presented, autoreactive B cells and autoantibodies represent ubiquitous pathomechanisms; findings obtained in the context of a specific disease will be transferable to others. Moreover, an increased understanding of pathomechanisms will enable the identification of promising new therapeutic approaches, preventing harm through pathogenic autoantibodies, but preserving beneficial effects of natural autoantibodies. As autoantibody-mediated processes are among the mechanisms that occur early in SSc, it is crucial to identify patients in the early stages of the disease, treat them with the listed therapeutics, and evaluate the potential to halt the development of full-blown disease. Ultimately, future technologies could enable the application of precision medicine in the treatment of B-cell-mediated pathologies: patient-specific autoantibodies mediating pathogenic effects could be identified enabling a patient-specific removal of autoantibodies. However, autoreactive B cells and autoantibodies represent only one field of the pathogenesis of SSc. An investigation of the intertwining of autoantibody-mediated pathomechanisms with other pathogenetic factors in SSc will be a major field of research in the future.

## Figures and Tables

**Figure 1 biomedicines-10-02150-f001:**
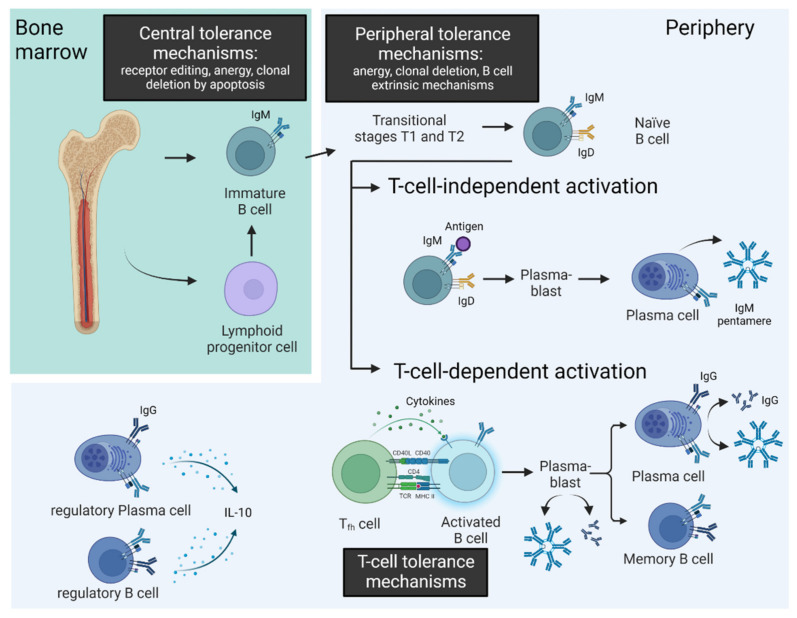
B-2 cell maturation in the bone marrow is characterized by the development of the BCR. Subsequently immature B cells leave the bone marrow. Naïve B cells can undergo T-cell-independent and T-cell-dependent activation. Tolerance mechanisms involving central tolerance mechanisms, peripheral tolerance mechanisms and T-cell tolerance mechanisms are marked in grey. The figure was created with ‘Biorender.com’.

**Figure 2 biomedicines-10-02150-f002:**
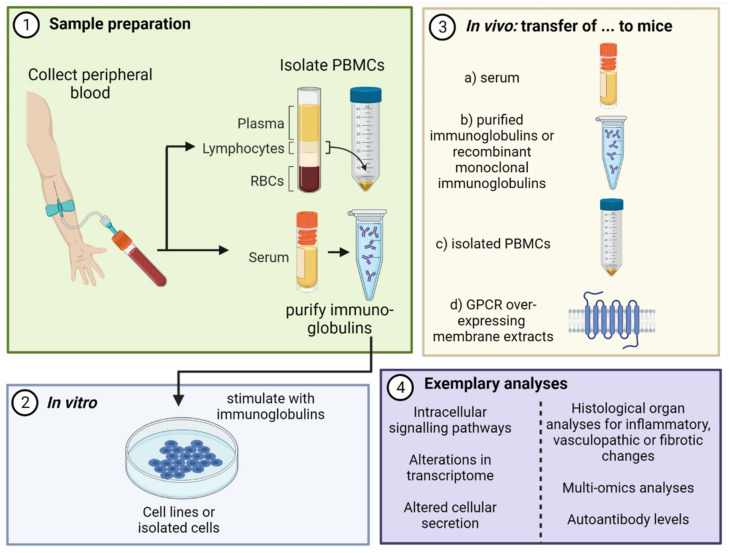
Established technologies to investigate functional effects of the autoantibodies in vitro and in vivo. The figure was created with ‘Biorender.com’.

**Figure 3 biomedicines-10-02150-f003:**
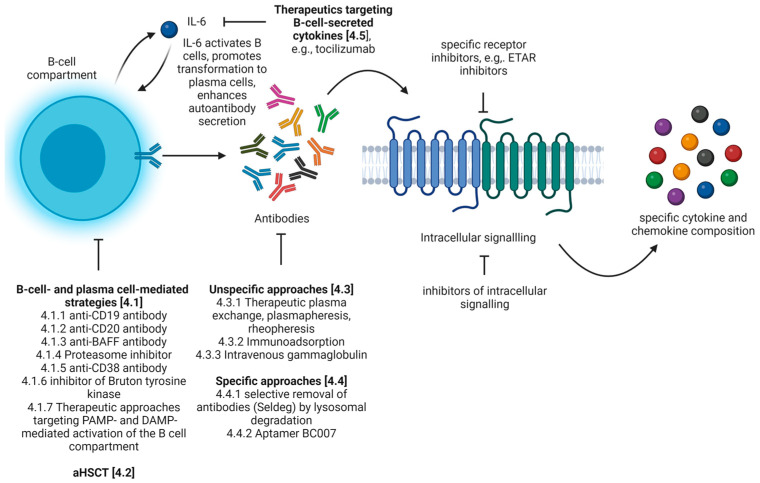
Potential therapeutic approaches for the treatment of autoantibody-induced pathologies in SSc. The figure was created with ‘Biorender.com’.

**Table 1 biomedicines-10-02150-t001:** Characteristics of natural and pathogenic autoantibodies.

	Natural Autoantibodies	Pathogenic Autoantibodies
Isotype	IgM, less frequently IgA or IgG	IgG, less frequently IgA or IgM
Generation of antibody diversity	Unmutated V(D)J recombination	V(D)J recombination, somatic hypermutation
Affinity	low	high
Mechanism of action	maintenance of immune homeostasis, amelioration of risk and severity of autoimmune diseases	contribution to autoimmune diseases via autoantibody-dependent and autoantibody-independent pathways

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
