# Peer review of "Autoantibodies as Biomarker and Therapeutic Target in Systemic Sclerosis"

_biomedicines, 2022, doi:10.3390/biomedicines10092150_

Round 1

Reviewer 1 Report

Very comprehensive review, congratulations. I only miss one thing: I suggest a supplemental chapter about danger signals (PAMP/DAMP) in the section about the B-cell activation: Toll-like receptors and their signal transduction play an important role in the pathogenesis and they could be probable targets in therapy too.

It was a review article, that means there is no any novelty in this manuscript. There are more than 10 papers about B-cells in scleroderma only in the last two years, which from 2 articles are very good review - so this manuscript is not new (https://pubmed.ncbi.nlm.nih.gov/35812378/, https://pubmed.ncbi.nlm.nih.gov/33745085/). However, this manuscript No 1820908 shows much more details about B-cells and their role in SSc than the other two, and more comprehensive too. So the scientific impact could be good in such topic which shows recently very high and always increasing interest in SSc-workgroups.

The main goal was a synthesis of all SSc-related topic in B-cell development, B-cell activation and antibody production. The authors are well known researchers in scleroderma-research, their opinion about importance of this topic is very determining. However, my personal sense/view is, that the authors wrote a little bit more examples about other diseases as it would be undoubting necessary. On the other hand, they are right, because of these examples could induce new original investigation in SSc very similarly and parallel than it was found in  other diseases.  The content of this manuscript was in all other aspects very relevant and interesting. I wrote in my original opinion too, that I see only one  topic which is failed in this paper: a summary about relations between DAMP signals and B-cells, mainly through the Toll-like receptors. This topic was not well explained in the other two reviews too - see above (only in the first one was one sentence about it). So I think, this supplementary data could be enhance the scientific impact of this manuscript and after this addition this manuscript will be more comprehensive than the foregoing articles were.

Is the paper well written? Is the text clear and easy to read?

Yes. It is little bit long and includes a lot of data, so the readers need more attention than for an original paper, but I think it is right for a review paper.  

Are the conclusions consistent with the evidence and arguments  presented? Do they address the main question posed?

Yes. The conclusion was short and wrote about an unmet need in SSc-research: we know about the role of B-cells in SSc pathogenesis (and therapy) very few  and the researchers should investigate this topics with more intensity.  

Author Response

Thank you for your helpful comments on the submitted manuscript. We appreciate the comments and have therefore made changes to the manuscript. One important recommendation related to the role of PAMP and DAMP in autoimmune diseases. We have therefore added lines 118-133. In addition, we have added section 3.1.7, which presents therapeutic aspects. Accordingly, we also adjusted Figure 3.
Another comment concerned the length and complexity of the manuscript. To focus the manuscript more on autoantibodies in systemic sclerosis, we revised the Introduction and Section 1. We have added Figures 1 and 2 and Table 1 to facilitate readability and highlight key aspects. We hope that the adjustments sufficiently address the reviewer's comments and improve the manuscript.

Reviewer 2 Report

This is an interesting review addressing systemic sclerosis (SSc) and discussing the alterations in the B cell compartment leading to polyclonal B-cell hyperreactivity, hypergammaglobulinemia and autoantibody production, the role of autoreactive B cells and autoantibodies in promoting and maintaining pathologic mechanisms. 

It is also discussed the role of autoreactive B cells and autoantibodies as potentially promising targets for therapeutic approaches including B-cell-targeting therapies, strategies for unselective and selective removal of autoantibodies, and presented mechanisms in the innate immune system leading to the generation of autoantibodies, alterations of the B cell compartment in SSc, autoantibodies as biomarkers and autoantibody-mediated pathologies in SSc and potential therapeutic approaches to target autoantibody-mediated pathologies.

The manuscript in general is well presented. However, regarding the diagnostic role of autoantibodies, some important literature data should be recalled.

-Autoantibodies against nuclear antigens (ANAs): the authors should further discuss the diagnostic role of autoantibodies to nuclear antigens and the established role of antibodies to extractable nuclear antigens (ENA). In particular, it has been reported that some ANA/ENA may be detected also in patients with other autoimmune diseases, such as primary biliary cholangitis (PBC), as serological markers of associated systemic sclerosis and may be useful for the early diagnosis of associated rhematological diseases, as previously reported (Antibodies to SS-A/Ro-52kD and centromere in autoimmune liver disease: a clue to diagnosis and prognosis of primary biliary cirrhosis. Aliment Pharmacol Ther. 2007 Sep 15;26(6):831-8.).

On the contrary, other ANA with different immunofluorescence patterns, such as "membranous" or "multiple nuclear dots" are very PBC specific, as previously demonstrated (Antinuclear antibodies giving the 'multiple nuclear dots' or the 'rim-like/membranous' patterns: diagnostic accuracy for primary biliary cirrhosis. Aliment Pharmacol Ther. 2006 Dec;24(11-12):1575-83.) thus further demonstrating the diagnostic relevance of ENA in PBC.

-Antineutrophil cytoplasmic antibodies (ANCAs): the authors mentioned only  anti-PR3 or MPO ANCA. They should, however, recall also ANCA detected by indirect immunofluorescence on neutrophils detected in autoimmune hepatitis, primary sclerosing cholangitis, and inflammatory bowel diseases which are classified as "atypical pANCA", as they are different from PR3 and MPO ANCA, as recently well discussed (Anti-neutrophil cytoplasm antibodies (ANCA) in autoimmune diseases: A matter of laboratory technique and clinical setting. Autoimmun Rev. 2021 Apr;20(4):102787. ).

Author Response

Thank you for the revision of the submitted manuscript. In accordance with the recommendations you made, we have included changes in the role of ANA and ANCA and an addition to the literature (see lines 320-322, 359-365, 392-395). We hope that the inserted changes will increase the quality of the submitted manuscript.

Round 2

Reviewer 2 Report

The authors satisfactorily addressed the raised points.